# Diagnosis of Cardiac Surgery-Associated Acute Kidney Injury: State of the Art and Perspectives

**DOI:** 10.3390/jcm11154576

**Published:** 2022-08-05

**Authors:** Alfredo G. Casanova, Sandra M. Sancho-Martínez, Laura Vicente-Vicente, Patricia Ruiz Bueno, Pablo Jorge-Monjas, Eduardo Tamayo, Ana I. Morales, Francisco J. López-Hernández

**Affiliations:** 1Institute of Biomedical Research of Salamanca (IBSAL), 37007 Salamanca, Spain; 2Unidad de Toxicología, Universidad de Salamanca (USAL), 37007 Salamanca, Spain; 3Department of Physiology and Pharmacology, Universidad de Salamanca (USAL), 37007 Salamanca, Spain; 4Group of Translational Research on Renal and Cardiovascular Diseases (TRECARD), 37007 Salamanca, Spain; 5National Network for Kidney Research REDINREN, RD016/0009/0025, Instituto de Salud Carlos III, 28029 Madrid, Spain; 6Department of Anesthesiology and Critical Care, Clinical University Hospital of Valladolid, 47003 Valladolid, Spain; 7Group of Biomedical Research on Critical Care (BioCritic), 47003 Valladolid, Spain; 8Centro de Investigación Biomédica en Red de Enfermedades Infecciosas (CIBERINFEC), Instituto de Salud Carlos III, 28029 Madrid, Spain

**Keywords:** cardiac surgery-associated AKI, diagnosis, biomarkers

## Abstract

Diagnosis of cardiac surgery-associated acute kidney injury (CSA-AKI), a syndrome of sudden renal dysfunction occurring in the immediate post-operative period, is still sub-optimal. Standard CSA-AKI diagnosis is performed according to the international criteria for AKI diagnosis, afflicted with insufficient sensitivity, specificity, and prognostic capacity. In this article, we describe the limitations of current diagnostic procedures and of the so-called injury biomarkers and analyze new strategies under development for a conceptually enhanced diagnosis of CSA-AKI. Specifically, early pathophysiological diagnosis and patient stratification based on the underlying mechanisms of disease are presented as ongoing developments. This new approach should be underpinned by process-specific biomarkers including, but not limited to, glomerular filtration rate (GFR) to other functions of renal excretion causing GFR-independent hydro-electrolytic and acid-based disorders. In addition, biomarker-based strategies for the assessment of AKI evolution and prognosis are also discussed. Finally, special focus is devoted to the novel concept of pre-emptive diagnosis of acquired risk of AKI, a premorbid condition of renal frailty providing interesting prophylactic opportunities to prevent disease through diagnosis-guided personalized patient handling. Indeed, a new strategy of risk assessment complementing the traditional scores based on the computing of risk factors is advanced. The new strategy pinpoints the assessment of the status of the primary mechanisms of renal function regulation on which the impact of risk factors converges, namely renal hemodynamics and tubular competence, to generate a composite and personalized estimation of individual risk.

## 1. Introduction

Acute kidney injury (AKI) is a syndrome of sudden renal excretory dysfunction, with high incidence and mortality, especially among the critically ill, and important long-term renal and cardiovascular morbidity [1]. AKI may be caused by a variety of factors, medical procedures, environmental conditions, drugs, and infections. Because AKI still lacks effective treatments other than supportive measures and withdrawal of the cause, an early and specific diagnosis is a yet unmet but critical need for successful and personalized patient handling [1,2,3]. Major surgery and, particularly cardiac surgery, is a considerable cause of in-hospital AKI, ranking second to sepsis in the context of intensive medicine [4]. Among critically ill patients, AKI has an extremely high mortality rate, reaching over 50% [5,6,7,8].

Post cardiac surgery-associated AKI (CSA-AKI) refers to AKI developing because of cardiac surgery, immediately following the procedure. Multifactorial CSA-AKI etiopathology encompasses featuring mechanisms, including cardiocirculatory deficit and renal embolisms, renal side effects of the extracorporeal circulation used during the operation (i.e., temporal renal ischemia, hemolysis, inflammation, and oxidative stress), and perioperative nephrotoxic drugs [9,10]. These mechanisms compromise renal hemodynamics and cause tubular damage, leading to the reduction in glomerular filtration rate (GFR) and the hydroelectrolytic disorders characteristic of AKI. CSA-AKI occurs in up to 43% of patients [10,11]. A recent systematic review and meta-analysis reported a composite incidence of 22.3% in adult cardiac surgery patients [12]. CSA-AKI is the strongest postoperative risk factor for death, with up to a 30% risk [13]. However, CSA-AKI diagnosis suffers from the same limitations as most forms of AKI, which limit patient handling and negatively impact prognosis. In this review, we critically analyze these limitations and describe the status of ongoing diagnostic developments including earlier and etiological diagnosis, and assessment of renal recovery and prognosis. We also make special emphasis on the novel concept of pre-emptive diagnosis of risk of AKI as a prospective tool to eventually help to prevent AKI with truly personalized criteria.

## 2. Diagnosis of CSA-AKI

### 2.1. The Standard Diagnosis of CSA-AKI Suffers from General AKI Diagnostic Criteria Limitations

Standard diagnosis of CSA-AKI is performed according to general international criteria for AKI diagnosis, the most recent of them being the KDIGO scale. The KDIGO scale, in addition to its predecessors (the RIFLE and AKIN scales) [1,14], is based on specified elevations of plasma creatinine concentration (Cr_pl_), a surrogate of GFR, in turn a surrogate of the ambiguous concept of “renal function,” or on reductions in urinary output. However, substantial limitations of standard criteria curtail the sensitivity and specificity of AKI diagnosis to suboptimal levels [1]. (1) Despite the multi-etiopathological nature of AKI, a one-fits-all, single biomarker (i.e., serum/plasma creatinine concentration, Cr_pl_)-based diagnosis is the international standard. This criterium neglects those forms and degrees of AKI not detected by this biomarker and provides no etiopathological granularity [15,16,17,18]. (2) Cr_pl_ is a late and cause-insensitive surrogate of GFR. Recruitment of renal functional reserve (RFR), i.e., increased single nephron GFR in undamaged nephrons, prevents the drop in overall GFR that would immediately follow and parallel renal damage (see Section 3.2.1). It is only after RFR-mediated compensation is overwhelmed that GFR starts to decrease and Cr_pl_ starts to increase. Consequently, the initial stages of severe AKI, in addition to all mild to moderate cases of AKI, pass unnoticed to Cr_pl_-based diagnosis [19,20].

### 2.2. A Widened Concept of AKI for an Earlier, More Sensitive and Etiological Diagnosis

From an etiopathological perspective, the two most common forms of AKI are the pre-renal (i.e., mostly hemodynamic, with no renal parenchymal damage) and the intrinsic (i.e., derived from primary renal tissue injury, most typically involving the tubule epithelium) types [21,22,23,24]. AKI patient handling and prognosis are conditioned by etiopathology and underlying pathophysiology. Pre-renal AKI evolves more favorably than intrinsic AKI, but alterations in Cr_pl_ are insensitive to this discrimination [15,16,17]. CSA-AKI (and AKI in general) may occur without renal damage (pre-renal), involving renal injury (intrinsic), but substantial damage may also occur with no alterations in GFR and Cr_pl_ [19,20]. Because this latter case is not classified as AKI by international scales, the term “subclinical AKI” was coined (Figure 1). This new concept was facilitated by the identification of “injury biomarkers” (mostly urinary) and became clinically relevant after it was noted that injury biomarker-positive, Cr_pl_-negative patients also had worsened prognosis [25,26,27]. Urinary “renal injury biomarkers” are typically represented by neutrophil gelatinase associated lipocalin (NGAL), kidney injury molecule-1 (KIM-1), interleukin 18 (IL18), N-acetyl-glucosaminidase (NAG), tissue inhibitor of metalloproteinases-2 (TIMP-2), insulin-like growth factor-binding protein 7 (IGFBP7), liver fatty acid binding protein (LFABP), and others [15,28,29,30,31,32].

Although not yet fully incorporated to international diagnostic criteria, a redefinition of AKI including injury biomarkers is increasingly installed, which recognizes three subcategories, as portraying increments in either Cr_pl_ or biomarkers, or in both [30,31,33,34,35]. In addition to affording early diagnosis [1,36,37,38], injury biomarkers facilitate pathophysiological sub classification of pre-renal and intrinsic AKI [1,39], as a complement to anamnestic classifications based on the response to fluid therapy, to biomarkers of tubular function (i.e., the fractional excretion of sodium—FENa–, and of urea –FEUr–, the urinary creatinine to plasma creatinine ratio—Cr_u_/Cr_pl_–, and others), and to the microscopic analysis of the urinary sediment [36,40,41,42,43,44,45]. For example, urinary calprotectin and NGAL have been reported to distinguish between pre-renal and renal AKI [36,39,46,47]. Similarly, NGAL but not KIM-1, is increased in human post-renal (i.e., obstructive) AKI [48]. These biomarkers are, however, potentially useful for the diagnosis of intrinsic forms of AKI, but the early, Cr_pl_-insensitive window of pre-renal cases is still bereft of indicators. Within this window, the altruism of competent nephrons (i.e., increased single nephron GFR -SNGFR-) conceals the underperformance of affected nephrons (those with reduced SNGFR), leading to a conserved overall GFR. This type of damage is hitherto impossible to detect with non- or minimally invasive technology.

A note of caution must be introduced about the clinical utility of injury biomarkers, as they constitute a rather ambiguous term and a heterogeneous class. First, the precise biological and clinical meaning of most (if not all) of these biomarkers is still insufficiently understood [49]. Second, the profile of urinary injury biomarkers is very heterogeneous among AKI patients, and among biomarkers, which indicates that each biomarker behaves distinctly and bears unique pathophysiological significance. This prevents their use as a class, and thus clinical application must be performed individually.

The pathophysiological complexity of syndromes affecting the renal excretory function include but supersede the alterations in mechanisms underpinning a reduction in GFR, as they also extend to tubular and interstitial injuries causing hydroelectrolytic and acid-base disorders unnoticed by current AKI definitions, but intimately ligated to it. Accordingly, conceptually new biomarkers of specific pathophysiological processes are needed to complement injury biomarkers, the latter of which are currently considered an ambiguous class with uncertain biological significance, based on statistical association with clinical outcomes. Further diagnostic refinement is under development to provide diagnostic granularity based on the assessment of renal blood flow in different renal zones, the extent and type of tubular cell death, the tubular segments affected, damage to the glomerular filtration barrier, interstitial infiltration, inflammation and cytokine profiles, functional reserve, glomerular filtration, tubular reabsorption, energetic and metabolic status, etc. For instance, Sasaki et al. [50] reported that a panel of urinary markers was able to detect the specific site of injury caused by different nephrotoxic drugs along the different nephron segments. Increasing precision on etiopathological diagnosis will impact patient management, as it will provide finer associations of a wider phenotypic spectrum with prognosis and treatment, according to the underlying damage pattern.

### 2.3. Assessment of Recovery and Prognosis

After a solved AKI episode, i.e., following Cr_pl_ normalization (or return to basal levels), patients are not usually monitored, and subclinical sequelae derived from ongoing or defective repair may pass largely unnoticed [51] (Figure 1). Subclinical sequelae are clinically relevant because patients with incomplete repair retain higher susceptibility to AKI recurrence [52], and misrepaired structures may eventually give way to progressive chronification of renal disease. Defective recovery from AKI impacts and causes short- and long-term morbimortality [53], whereas effective recovery is associated to lower risk of long-term mortality and adverse renal complications [53,54]. For instance, AKI increases mortality odds in the first week, even after renal function normalization [55], and lingering AKI may evolve to acute kidney disease (AKD) eventually leading to chronic kidney disease (CKD) [56,57] in 19–31% of the cases [58]. Even reversible AKI is associated with augmented incidence of CKD [59]. End-stage renal disease caused by unsolved AKI raised from 1.2% by 1998 to 1.7% by 2003 and will continue to scale due to aging and increase in comorbidity [58].

The concept and profiling of AKI recovery are still incompletely defined. The Acute Dialysis Quality Initiative (ADQI) group generated a consensus definition [60] based on the improvement of Cr_pl_, which perpetuates the diagnostic limitations of this biomarker [61,62]. Consequently, during AKI recovery, Cr_pl_ may return to normal levels prior to complete GFR and structural restoration. Thus, inclusion of structural restoration assessment is necessary for monitoring the subclinical sequelae of AKI. Therefore, Kashani and Kellum [63] suggested to include injury biomarkers in the definition of recovery, a strategy constrained by their limitations, as outlined in Section 2.2. Individual studies have indicated that KIM-1 and the furosemide stress test (FST; see Section 3.3.2) are sensitive and correlate with the lingering subclinical sequelae of AKI. In particular, after the normalization of the standard markers of glomerular filtration and tubular function, persistent histological injury correlates with the renal expression and urinary level of KIM-1 in rats [64]. Additionally, the response to the FST is not only sensitive to extant subclinical tubular injury, but also correlates with the persistence of a state of increased risk of new episodes of AKI [52]. Both markers are interesting candidates to be prospectively converted into diagnostic tools for the non-invasive follow-up of renal repair.

Advanced complications of AKI sequelae—for instance, progression from AKI to CKD—are also of evident diagnostic interest. Skewed processes underlying misrepair, fibrosis, and renal parenchyma structural derangement may yield new biomarkers [65]. However, caution should be taken with respect to the usefulness of these markers because these processes are also common to CKD patients, so they might not differentiate pre-existing CKD from the AKI-to-CKD transition when renal status is ignored prior to the AKI episode. Indeed, whilst NGAL and KIM-1 mark AKI-to-CKD [63,66], both markers [67,68,69,70,71,72,73,74], and also LFABP [67,75], have been reported to be elevated in patients with stablished CKD. NGAL also predicts poor outcomes in CKD patients (i.e., progression to end-stage renal disease) [76,77]. Accordingly, all these markers may have important limitations to detect AKI-to-CKD transition, as their level may simply reflect AKI or CKD.

## 3. Pre-Emptive Assessment of Risk: Perspectives of a New Diagnostic Concept for CSA-AKI

In the last decade, the concept of risk of AKI has emerged as a premorbid condition of frailty predisposing to disease (Figure 1). Yet, the condition of risk provides a window of opportunity to prevent progression to disease in predisposed patients and to apply with confidence beneficial, but risks potentially kidney-injuring procedures and treatments to non-predisposed patients. However, diagnosis of predisposition to AKI is still under development, and poses an immediate challenge for preventive and precision medicine. Risk may be underpinned by genetic and acquired factors. Genetic predisposition to AKI has been recently reviewed [78]. Despite that several gene polymorphisms have been associated to AKI risk, further understanding of the genetics predisposing to AKI is still necessary to impact prevention and treatment. In this section, the novel concept of acquired predisposition to AKI is addressed. Acquired predisposition refers to a condition of renal frailty eventually caused by risk factors including environmental factors (such as drugs, medical procedures, toxins, etc.) and attendant comorbidities affecting the efficacy of the physiological regulation network governing the defense of renal function homeostasis and renal tissue integrity. As opposed to genetic predisposing factors, risk factors underpinning acquired predisposition are largely modifiable for prophylactic intervention. Diagnosis of risk status is thus critical for patient triage and stratification.

### 3.1. From Risk Factor-Based to Personalized Assessment of Risk

Risk factors for CSA-AKI have been identified (Table 1) based on which risk scores have been developed (Table 2). These scores provide reported diagnostic accuracy in between 67 and 89%, mostly from retrospective studies. Larger, multicentric and prospective studies are necessary to further explore their validity under different clinical scenarios and circumstances. Importantly, risk factor-based scores do not afford truly personalized diagnosis, as they project population probability statistics on the individual. As a result, all individuals bearing a specific risk factor are assigned the same odds of undergoing CSA-AKI, whereas in practice only a fraction of them will develop the syndrome.

We contend that increasing diagnostic precision should be attained by individually assessing the impact of risk factors on the direct determinants of renal function, rather than only computing their presence or absence (Figure 2). This is because (i) each risk factor impacts distinctly on each patient; (ii) accumulation of risk factors does not necessarily alter additively the determinants of renal function; (iii) non-evident risk factors may pass unnoticed but still impact risk. Conversely, it is the actual status of the renal function regulation network, as distinctly conditioned in each patient by the specific combination of known and unknown risk factors, that determines individual risk and thus the object of assessment for complementary diagnosis [95]. Specifically, as for most forms of AKI, pathological mechanisms activated by cardiac surgery funnel toward renal hemodynamics and tubular performance. Perioperative drugs, impaired cardiac competence, inflammation-derived vascular dysfunction, and embolisms may cause renal hemodynamic deficit; whilst drugs, and extracorporeal circulation-associated inflammation, ischemia and hemolysis may cause tubular injury, leading to pre-renal (i.e., hemodynamic) and intrinsic AKI, respectively (Figure 3) [4,9,10,11,96]. A corollary follows that hemodynamically frail patients and those with subclinical tubular alterations may be predisposed to CSA-AKI. Thus, evaluation of hemodynamic and tubular status emerges as core aspects of risk assessment.

### 3.2. Assessment of Renal Hemodynamic Frailty

Hemodynamic frailty (HDF) has been recently described as a state of increased predisposition to disease characterized by exhaustion or partial limitation of the haemodynamic reserve and adaptive haemodynamic responses [95]. HDF defines the handicapped ability to maintain tissue perfusion and volemia in response to challenges such as fluid imbalance while, in the absence of stressors, basal haemodynamic competence is achieved. As in highly autoregulated vascular beds, renal haemodynamic homeostasis is largely dependent on renal autoregulation, the mechanism maintaining GFR (and renal blood flow, RBF) constant upon fluctuations in blood pressure, within an ample range (Figure 4). Reductions in renal perfusion pressure (i.e., systemic blood pressure) or cardiac output do not translate into reductions in renal blood flow or net glomerular filtration pressure (i.e., the direct determinants of GFR), unless renal autoregulation performance is impaired. However, as indicated in Figure 4, impaired renal autoregulation narrows the autoregulation range [97,98,99] and pronounces the impact of variations in systemic blood pressure on renal haemodynamics.

Renal autoregulation is bestowed by contraction and dilation of the efferent, but primarily of the afferent glomerular arterioles as a response to variations in blood pressure (and in GFR). When blood pressure increases, afferent arterioles contract to prevent the increment in blood pressure to translate to the glomerulus, and when blood pressure drops, the opposite occurs resulting, in both cases, in GFR stabilization. Thus, dysfunction or exhaustion of the autoregulation capacity handicaps regulation responses and thus poses the key determinant of renal hemodynamic frailty and predisposition to AKI. 

#### 3.2.1. Assessment of Renal Functional Reserve as a Surrogate for Renal Hemodynamic Frailty

The availability of renal autoregulation led to the concept of renal functional reserve (RFR). Renal functional reserve has been defined as the capacity of the kidney to increase glomerular filtration rate (GFR) in response to certain physiological or pathological stimuli or conditions [100], which critically depends on functional autoregulation. In fact, RFR is recruited by dilation of the afferent arterioles, so that available RFR may be taken as a surrogate of functional autoregulation. 

Applied to renal pathology, RFR is thought to be recruited to elevate single nephron GFR (SNGFR) in remaining nephrons during acute or chronic conditions characterized by nephron loss or nephron functional loss. In such scenarios, increased SNGFR would maintain overall GFR [100,101,102], implying that RFR must be exhausted before the overall GFR declines. Unavailable RFR, as by exhaustion or irresponsiveness, causes a state of vulnerability and predisposition to AKI that sets the individual on the verge of dysfunction [1,103,104]. Because the RFR can be recruited by an amino acid bolus or a protein-rich meal [105,106], a functional RFR test exists as the difference between the GFR in the resting state and at maximum capacity, which has been widely used in multiple studies and is proposed as a novel renal biomarker [107,108]. Consequently, the RFR test might be used to probe renal hemodynamic frailty and risk of hemodynamic AKI.

#### 3.2.2. Assessment of Dehydration: A Relevant Inducer of Renal Hemodynamic Frailty

Dehydration poses a crucial determinant of hemodynamic frailty that predisposes a patient to both pre-renal and intrinsic AKI [95]. Common in both types of AKI (as potentially occurring following cardiac surgery) is the reduction of GFR resulting from a lower RBF and net filtration pressure (Figure 5). Dehydration precipitates vegetative responses pursuing the maintenance of blood pressure, tissue perfusion, and GFR. Within the kidneys, these responses primarily converge on afferent contraction (and reduction of RBF) and Na+ sparing effects. Afferent contraction amplifies the effect of incidental renal insults by setting the basal pathophysiological scenario under compromised perfusion, and in a steeper position of the RBF to GFR relationship (Figure 6). In these circumstances, RBF-lowering insults, such as those triggering both pre-renal and intrinsic AKI, would cause a greater GFR reduction than if they acted from a less steep segment of the curve. 

Dehydration is particularly prevalent in populations at higher risk of AKI, in which this syndrome has devastating consequences, particularly in older individuals [109,110] and critically ill patients [111]. Among the aged, dehydration is commonly found in 20–30% of the individuals [112], and over 50% of individuals aged 65 years or older are actively or at stake of becoming dehydrated [113]. Assessment of dehydration as a component of risk of AKI might also be advised prior to cardiac surgery. From the pathophysiological and clinical points of view, dehydration is a very complex condition with both hypertonic and isotonic presentations and manifold implications, for whose definition, measurement, and handling there is no universal consensus. Yet, the gold standard diagnostic method is the direct measurement of plasma/serum osmolality, a rather unreliable parameter with only gross sensitivity [114]. Electrical bioimpedance studies and other non-invasive technologies are also available.

### 3.3. Assessment of Subclinical Tubular Competence

Tubular competence may be compromised by lethal (i.e., involving tubular cell death) and sub-lethal alterations, potentially causing hydroelectrolytic disorders and GFR to decrease (Figure 5). In addition, a sub-lethal, subclinical, and injury biomarker-negative, acquired condition predisposing to intrinsic (i.e., tubular) AKI has been described in animal models [103,115,116,117,118]. In these studies, animals were predisposed by sub-nephrotoxic regimes of potentially nephrotoxic drugs (e.g., cisplatin or gentamicin) or toxins (e.g., uranium). Predisposed animals indicated normal renal histology, normal standards of renal function (including normal GFR and Cr_pl_), normal parameters of tubular performance (such as the fractional excretion of Na+), and unaltered urinary excretion of injury biomarkers (e.g., NGAL, KIM-1, NAG). When subject to a stressor (e.g., another drug) that caused no harm to non-predisposed animals, predisposed animals developed an overt AKI in the form of severe acute tubular injury. Interestingly, these models have been useful to identify biomarkers of predisposition, as described in the following sections.

#### 3.3.1. Urinary Biomarkers of Predisposition to Tubular AKI

When measured before, during, or immediately after cardiac surgery, the urinary level of some of the well-stablished as injury biomarkers associate with post-operative CSA-AKI. Although further clinical research is necessary, they might be eventually transformed into predictive diagnostic tools. For instance, several studies have indicated that the urinary level of NGAL excreted intraoperatively and after surgery is effective in predicting AKI in both neonates and children, and in adults, with higher efficacy in the former [119,120,121,122]. Similar results have been obtained with other urinary biomarkers, such as the cell cycle arrest biomarkers TIMP-2 and IGFBP7) [120,123,124], KIM-1 [121,125,126], NAG [121], IL-18 [121,127], and L-FABP [121,126,128], both alone and in combinations.

Additionally, new urinary biomarkers associated to predisposition to AKI have been identified in animal models, including ganglioside M2 activator protein (GM2AP) [118], fumarylacetoacetase (FAA) [116], hemopexin [115], transferrin [115,117], albumin [115,129] and the vitamin D binding protein (VDBP) [115]. Many of these biomarkers were model-specific, and thus their potential clinical application was restricted to determined scenarios. However, transferrin demonstrated a wider application range [117]. Interestingly, urinary transferrin proved to be reflective of the predisposition to AKI induced by subclinical tubular alterations regardless of etiology. In this study, the urinary level of transferrin prior to subjecting animals to a triggering insult tightly associated with the subsequent AKI in those animals in which risk had been induced by insults to the renal tubules (i.e., cisplatin, gentamicin, and uranyl nitrate). Conversely, transferrin indicated no relation with the risk posed by cyclosporine A, a drug known to cause AKI by altering renal hemodynamics. Congruently, increases in the urinary excretion of transferrin during the risk state were demonstrated to result from a reduction in its tubular reuptake, supporting a potential diagnostic role based on a pathological mechanism (i.e., tubular affection) rather than on specific etiologies.

Furthermore, urinary transferrin provided significant accuracy at pre-emptively identifying oncological and cardiac patients at risk of AKI in two recent clinical studies [117,129]. In addition to transferrin, albuminuria and GM2AP indicated statistically significant predictive capacity in one of these studies [129]. Indeed, albuminuria demonstrated the highest diagnostic power, with a cutoff threshold established within the subclinical rage (10–30 µg/mg urinary Cr). Altogether, urinary transferrin, albumin, and GM2AP seem to be interesting candidates for prospective diagnostic methods aimed at predicting AKI in patients to be subject to cardiac surgery, which must be more deeply studied in larger clinical studies.

#### 3.3.2. The Furosemide Stress Test

The characteristics of the FST make it an outstanding candidate for the detection of subclinical tubular alterations and, thus, for the diagnosis of predisposition to AKI, including CSA-AKI. Recently standardized to study tubular function integrity in humans [130,131], and more recently in rats [52,103], the FST consists of the measurement of the acute, timed diuretic and kaliuretic response to a single administration of furosemide. This response is dependent on intact secretion of furosemide to the lumen of the proximal tubule from peritubular capillaries, and the binding and inhibition of the NKCC transporter in the loop of Henle [132]. Tubular alterations causing impaired proximal secretion of furosemide or action at the loop of Henle result in reductions in the diuretic and kaliuretic response. Because of the specific kinetic features and NKCC inhibition properties, the FST is a high-sensitivity but low-resolution test that detects subtle alterations in tubular performance, but is less sensitive to distinguishing dysfunctional grade or the tubular segments affected [103]. Very interestingly, the FST has been indicated to detect subclinical conditions such as predisposition to AKI [103] and the sequelae persisting beyond Cr_pl_ normalization after AKI [52], which include a higher risk of additional AKI episodes. This makes the FST a suitable candidate biomarker for the diagnosis of risk of CSA-AKI.

In perspective, the information provided by tests and markers of renal hemodynamic frailty and subclinical tubular performance may be interpreted individually but, optimally, a computerized system calculating risk based on an integral evaluation of the pathophysiological status of the entire renal regulation network must be eventually generated.

## 4. Conclusions

Present diagnosis of CSA-AKI suffers from suboptimal specificity and sensitivity. New strategies are under development for an earlier and etiological diagnosis, and for the assessment of recovery and prognosis following AKI, which might be prospectively useful for CSA-AKI integral management. Most notably, the novel concept of pre-emptive estimation of risk of AKI may eventually help to prevent CSA-AKI. Pre-emptive risk assessment is proposed to be provided by artificial intelligence-assisted, individual evaluation of the status of the primary determinants of renal function, namely renal hemodynamics and tubular homeostasis with precision medicine criteria. This strategy is expected to complement or supersede present risk assessment based on the indiscriminate projection of group statistics on the individual.

## Figures and Tables

**Figure 1 jcm-11-04576-f001:**
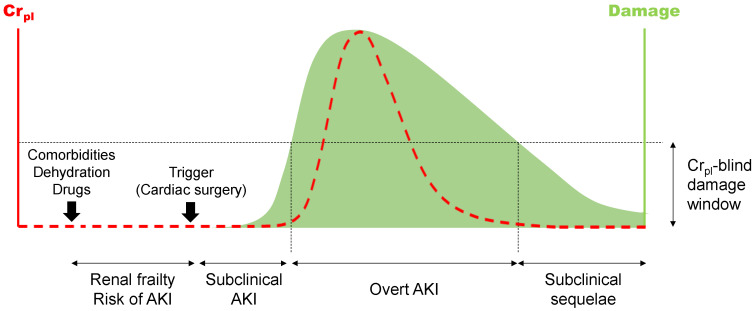
Stages of AKI potentially subject to diagnosis. AKI, acute kidney injury. Cr_pl_, plasma creatinine concentration.

**Figure 2 jcm-11-04576-f002:**
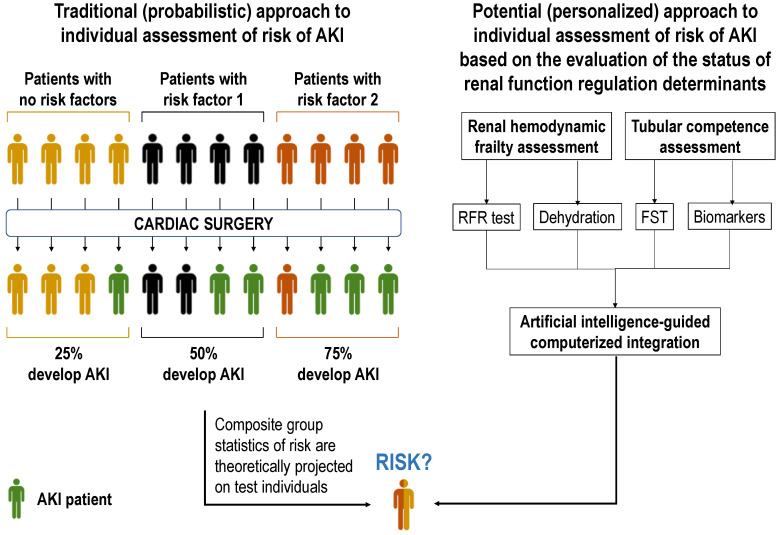
Schematic representation of two approaches to CSA-AKI risk estimation: a traditional, probabilistic approach based on the scoring of risk factors (**left**), and a new, potential approach based on the assessment of the impact of risk factors on the direct determinants of renal function (**right**). AKI, acute kidney injury. FST, furosemide stress test. RFR, renal functional reserve.

**Figure 3 jcm-11-04576-f003:**
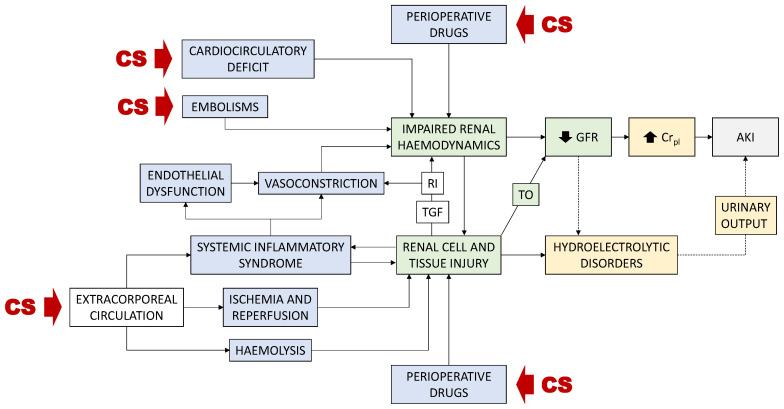
Pathophysiological mechanisms of CSA-AKI. AKI, acute kidney injury. Cr_pl_, plasma creatinine concentration. CS, cardiac surgery. GFR, glomerular filtration rate. RI, renal inflammation. TGF, tubuloglomerular feedback. TO, tubular obstruction.

**Figure 4 jcm-11-04576-f004:**
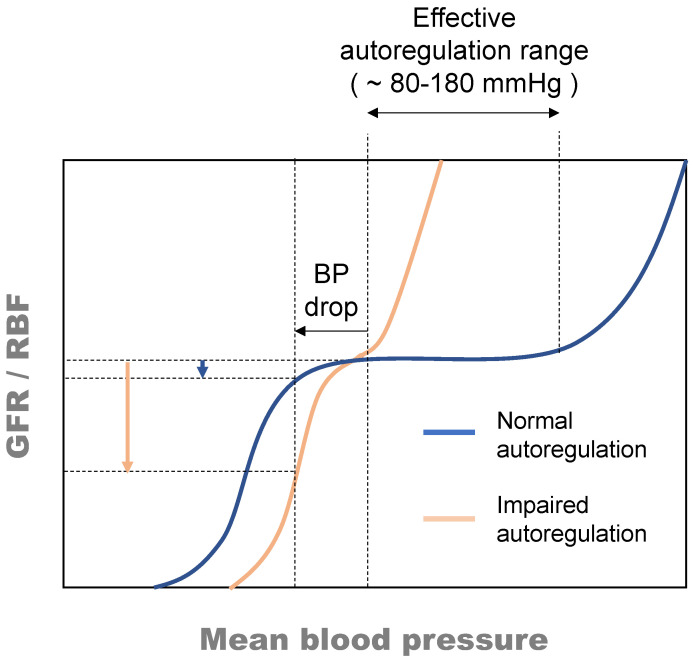
Relation between blood pressure (i.e., renal perfusion pressure) and glomerular filtration rate or renal blood flow under normal and impaired autoregulation conditions indicating that, under impaired autoregulation, the effect of a drop in blood pressure (BP) on glomerular filtration rate (GFR) or renal blood flow (RBF) is magnified, compared to the effect observed under normal autoregulation conditions [95].

**Figure 5 jcm-11-04576-f005:**
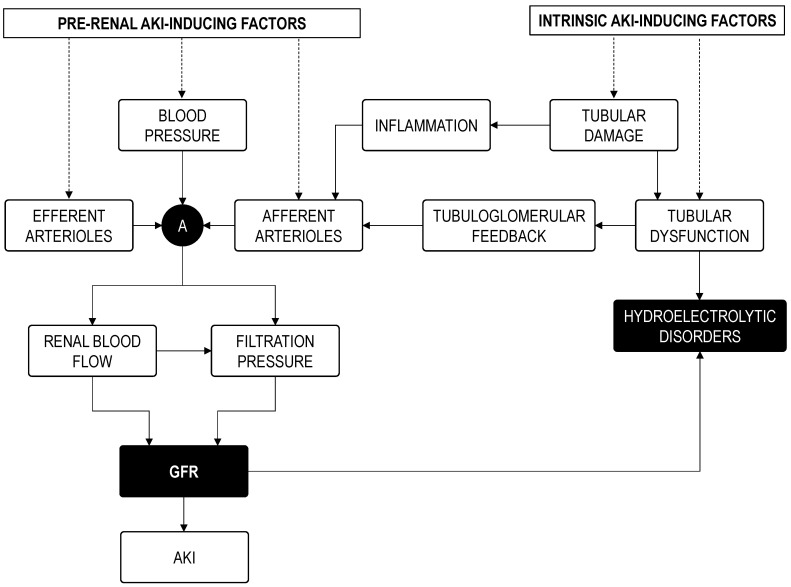
Simplified depiction of the pathophysiological scenario explaining the confluence of pre-renal AKI-inducing mechanisms and intrinsic AKI-inducing mechanisms at reducing glomerular filtration rate (GFR). A, autoregulation mechanisms. AKI, acute kidney injury.

**Figure 6 jcm-11-04576-f006:**
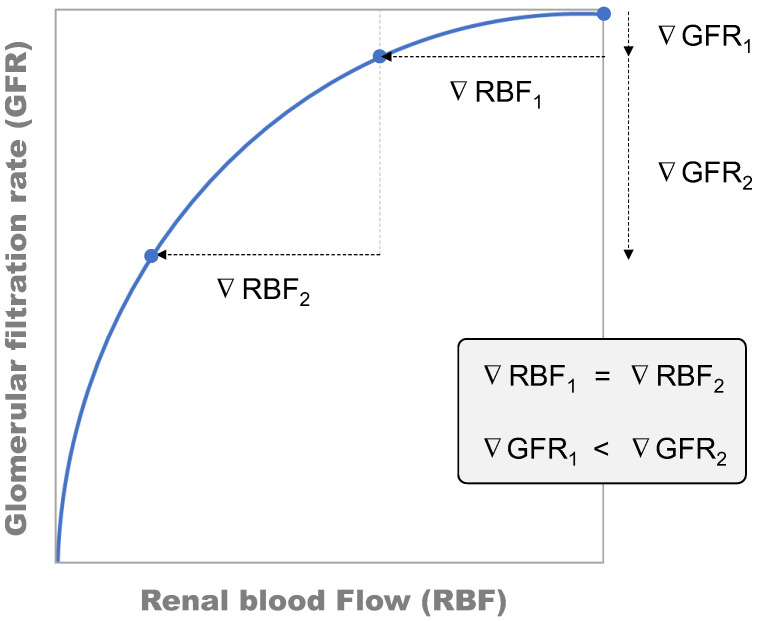
Effect of renal blood flow (RBF) on glomerular filtration rate (GFR) demonstrating a parabolic relationship where the reduction of GFR caused by a reduction in RBF depends on the starting point [95].

**Table 1 jcm-11-04576-t001:** Primary risk factors for CSA-AKI. CPB, cardiopulmonary bypass; NSAID, non-steroidal anti-inflammatory drug; NYHA, New York Heart Association Scale. Adapted from [4,10,79,80].

	Inherent to the Patient	Related to the Surgical Procedure
Preoperative	Female sex Advanced age Severe cardiac disease Previous cardiac surgery Congestive cardiac failure NYHA class III/IV Left ventricular ejection fraction <35% Left main coronary artery disease Peripheral vascular disease Hypertension Anemia Generalized atherosclerotic disease Chronic obstructive pulmonary disease Previous cerebrovascular accidents Diabetes mellitus Respiratory system disease Chronic kidney disease Chronic liver disease Consumption of nephrotoxic drugs, i.e., angiotensin-converting enzyme inhibitors, angiotensin receptor blockers, diuretics, or NSAIDs.	Preoperative cardiac angiography Preoperative insertion of intra-aortic balloon pump Need for emergency surgery Complex surgical procedure Emergency surgical procedure
Intraoperative		Type of surgery Duration of CPB (>100–120 min) CPB non-pulsatile Aortic clamping time Low mean arterial pressure during CPB Hypothermic CPB Deep hypothermic circulatory arrest Perioperative hemodilution Perioperative red blood cell transfusion Haemolysis and haemoglobinuria Embolism Iodinated contrast media
Postoperative		Low cardiac output Hypotension Intense vasoconstriction Atheroembolism Sepsis Use of norepinephrine Administration of nephrotoxic drugs (i.e., diuretics such a furosemide and tiazides, NSAIDs such as ibuprofen and acetylsalicylic acid, or aminoglicoside antibiotics, such as gentamicin and amikacyn). Infection Redo operation

**Table 2 jcm-11-04576-t002:** Risk scores for the prediction of CSA-AKI based on the algorithmic computation of risk factor. AUC, area under the curve; BMI, body mass index; CABG, coronary artery bypass grafting; CCB, calcium channel blockers; CHD, coronary heart disease; CHF, chronic heart failure; CoHF, congestive heart failure; CKD, chronic kidney disease; CPB, cardiopulmonary bypass time; CrCl, creatinine clearance; CVP, central venous pressure; GFR, glomerular filtration rate; IABP, intraaortic balloon pump; LDH, lactate dehydrogenase; NSAID, nonsteroidal anti-inflammatory drugs; NYHA, New York Heart Association Scale; PPI, proton pump inhibitors; PVD, peripheral vascular disease; SMA, simplified model approximation.

Reference	Center, Location	Number of Patients	Diagnostic Accuracy (AUC/c-Statistic)	Risk Factors Included
Chertow et al., 1997 [81]	43 Department of Veterans Affairs medical centers, USA	42,773	0.76	- Valvular surgery. - Decreased CrCl. - Intra-aortic balloon pump. - Prior heart surgery. - NYHA class IV. - Peripheral vascular disease. - Ejection fraction <35%. - Pulmonary rales. - Chronic obstructive pulmonary disease. - Elevated systolic blood pressure with coronary artery bypass graft surgery.
Thakar et al., 2005 [82]	Cleveland Clinic Foundation, USA	15,838	0.81	- Female gender. - Congestive heart failure. - Left ventricular ejection fraction < 35%. - Preoperative use of intra-aortic balloon pump. - Chronic obstructive pulmonary disease. - Insulin-requiring diabetes. - Previous cardiac surgery. - Emergency surgery. - Valve surgery only (reference to CABG). - CABG + valve (reference to CABG). - Other cardiac surgeries. - Elevated preoperative creatinine.
Mehta et al., 2006 [83]	STS National Cardiac Surgery Database, USA and Canada	444,524	0.84/0.83 (SMA)	- Decreased glomerular filtration rate. - Elevated serum creatinine (only SMA). - Aortic valve surgery (SMA). - Aortic valve surgery plus CABG (SMA). - Mitral valve surgery (SMA). - Mitral valve surgery plus CABG (SMA). - Age (in 5-year increments starting at age < 50 years) (SMA). - Diabetes treated with insulin (SMA). - Diabetes treated with oral agents (SMA). - Chronic lung disease (SMA). - Myocardial infarction in last 3 weeks (SMA). - Cardiogenic shock (SMA). - NYHA class IV (SMA). - Race (nonwhite vs white) (SMA). - Prior CV surgery (SMA). - Female. - Peripheral or cerebrovascular disease. - Body surface area. - Left ventricular ejection fraction. - Emergent status, salvage/resuscitation vs elective/urgent. - Emergent status, emergent (no salvage) vs elective/urgent. - Triple-vessel disease. - Left main disease. - Prior percutaneous coronary interventions. - Hypertension. - Immunosuppressive treatment. - Aortic stenosis. - Mitral insufficiency.
Wijeysundera et al., 2007 [84]	2 hospitals in Ontario, Canada	10,751	0.81	- Decreased GFR. - Diabetes mellitus requiring medication. - Left ventricular ejection fraction <40%. - Previous cardiac surgery. - Procedures other tan isolated coronary artery bypass graft or isolated atrial septal defect repair. - Nonelective procedure. - Preoperative intra-aortic balloon pump.
Aronson et al., 2007 [85]	Seventy institutions in 17 countries (Multicenter Study of Perioperative Ischemia)	2381	0.84	- Age >75 years. - Congestive heart failure. - Myocardial infarction. - Renal disease. - Inotropes. - Intra-aortic balloon pump. - CPB time ≥122 min. - Elevated pulse pressure.
Palomba et al., 2007 [86]	Heart Institute, University of São Paulo, Brazil	603	0.84	- Combined surgery. - CHF NYHA >2. - Pre-operative creatinine >1.2 mg/dL. - Low cardiac output. - Age >65 years old. - CPB time >120 min. - Pre-operative capillary glucose >140 mg/dL. - CVP >14 cm H_2_O.
Brown et al., 2007 [87]	8 medical centers in Vermont, New Hampshire, and Maine, New England, USA	8363	0.72	- Advanced age. - Female. - Diabetes. - PVD. - CoHF. - Hypertension. - Prior CABG surgery. - Preoperative IABP. - White blood cell count >12,000.
Heise et al., 2010 [88]	University Hospital of Goettingen, Germany	3508	0.67	- Elevated preoperative creatinine. - Preoperative use of IABP. - Emergency surgery. - Insulin requiring diabetes. - Female gender. - Cerebrovascular disease.
Jorge-Monjas et al., 2016 [89]	Clinic University Hospital, Valladolid, Spain	810	0.89	- Elevated creatinine. - Long CPB time. - Elevated lactate. - High EuroSCORE.
Guan et al., 2019 [90]	The Affiliated Hospital of Qingdao University, China	1900	0.80	- Decreased GFR. - Surgery history. - Elevated LDH. - Antibiotic use. - Age. - Long prothrombin time. - CHD. - CCB use. - PPI use. - Transfusion. - Cardiac arrhythmia. - CKD. - NSAID use. - Statin use.
Che et al., 2019 [91]	Shanghai Tongren Hospital and Clinical Research Institute, Shanghai, China	2552	0.80	- Advanced age. - Hypertension. - Previous cardiac surgery. - Hyperuricemia. - Prolonged operation time. - Postoperative central venous pressure <6 mm H_2_O. - Low postoperative cardiac output.
Callejas et al., 2019 [92]	23 hospitals in Spain	942	0.72	- Anemia. - Valve surgery ± CABG. - Other cardiac surgeries. - Age ≥70 years. - Congestive heart failure. - Previous cardiac surgery. - BMI ≥30. - Hypertension.
McBride et al., 2019 [93]	Cardiac Surgical Unit of the Royal Victoria Hospital, Belfast, UK	344	Not calculated	- Age ≥65 years. - High BMI. - Diabetes. - CPB time ≥130 min. - Cross clamp time ≥90 min. - Operation time ≥296 min. - Intra-aortic balloon pump. - Packed red blood cells. - Platelet bags. - Resternotomy.
Coulson et al., 2021 [94]	33 hospitals from Australia and New Zealand	22,731	0.68 (preoperative score)/0.70 (post-operative score)	- Preoperative score: - Preoperative hemoglobin <130 g/L. - Preoperative creatinine >100 µmol/L. - Age >70 years. - NYHA status 4. - BMI >30. - Post-operative score: - Preoperative hemoglobin <120 g/L. - Preoperative creatinine >100 µmol/L. - Perfusion time >100 min. - NYHA group 4. - BMI >30.

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
