# Peer review of "Diagnosis of Cardiac Surgery-Associated Acute Kidney Injury: State of the Art and Perspectives"

_jcm, 2022, doi:10.3390/jcm11154576_

Round 1

Reviewer 1 Report

In the present review the authors describe the limitations of current diagnostic procedures and of the renal injury biomarkers and analyze new strategies under development for a conceptually enhanced diagnosis of cardiac surgery-associated acute kidney injury (CSA-AKI). Furthermore, biomarker-based strategies for the assessment of AKI evolution and prognosis are also discussed. Finally, the authors focus to the novel concept of pre-emptive diagnosis of acquired risk of AKI, which provide opportunities to prevent disease by using diagnosis-guided personalized patient handling.

Overall the review is well organized and presents interesting data.

Comments for minor revision:

1. The Introduction needs further work. In particular, the authors should mention why their review is interesting by highlighting the main topics to be discussed in the article.

2. The authors should discuss in brief the prevention of CSA-AKI by risk stratification using urinary TIMP-2 and IGFBP7 (Biomark Med. 2021 Oct;15(14):1201-1210. doi: 10.2217/bmm-2020-0656).

3. Please provide the most important nephrotoxics in Table 1.

4. The authors should add a Conclusion section to the manuscript summarizing the key points of the review and future perspectives.

Author Response

In the present review the authors describe the limitations of current diagnostic procedures and of the renal injury biomarkers and analyze new strategies under development for a conceptually enhanced diagnosis of cardiac surgery-associated acute kidney injury (CSA-AKI). Furthermore, biomarker-based strategies for the assessment of AKI evolution and prognosis are also discussed. Finally, the authors focus to the novel concept of pre-emptive diagnosis of acquired risk of AKI, which provide opportunities to prevent disease by using diagnosis-guided personalized patient handling.

Overall the review is well organized and presents interesting data.

Authors: We thank the Reviewer for their positive consideration of our article and for the useful comments to improve the manuscript.

Comments for minor revision:

  1. The Introduction needs further work. In particular, the authors should mention why their review is interesting by highlighting the main topics to be discussed in the article.

Authors: Yes, certainly. Thank you for spotting this deficiency, as well as the missing conclusion referred to in the Reviewer’s comment 4, below. The article will be much better finished with a proper justification and conclusions.

  1. The authors should discuss in brief the prevention of CSA-AKI by risk stratification using urinary TIMP-2 and IGFBP7 (Biomark Med. 2021 Oct;15(14):1201-1210. doi: 10.2217/bmm-2020-0656).

Authors: While prevention is, of course, a final goal of pre-emptive diagnosis, it is really out of the scope of this article. There are many potential preventive strategies based (or not) on pre-emptive diagnosis that would be worth to be addressed, should this concept be among the article’s goals. We do not quite much see a clear point as to mention only a specific one.

  1. Please provide the most important nephrotoxics in Table 1.

Authors: We have added a shortlist of the most important nephrotoxics impacting renal function in the context of CSA.

  1. The authors should add a Conclusion section to the manuscript summarizing the key points of the review and future perspectives.

Authors: Following with our response to Reviewer’s comment 1, we have added a Conclusions section at the end of the manuscript. Thank you.

Reviewer 2 Report

The review article entitled "Diagnosis of cardiac surgery-associated acute kidney injury: State of the art and perspectives" talks about the development and future directions of diagnosing CSA-AKI. The tables, figures, and graphs provided along with the article are very thorough and informative. However, the authors should address the below concerns/comments to improve the quality of the article.    

1. Authors should include a couple of sentences in the Introduction highlighting the factors responsible for AKI's pathogenesis due to cardiac surgery. It will help readers who are not experts in the field.

2. Diagnosis of CSA-AKI appears to be the same as the diagnosis of AKI in general. Is there any biomarker or pathophysiological parameters that can directly indicate the involvement of cardiac surgery in AKI?

3. Do the authors think that the furosemide stress test could be specific for detecting CSA-AKI? can it not mislead in finding the real cause of kidney failure if considered as specific to CSA-AKI biomarker?  

4. A summary/conclusion section is required to summarize the review in the last.

Author Response

The review article entitled "Diagnosis of cardiac surgery-associated acute kidney injury: State of the art and perspectives" talks about the development and future directions of diagnosing CSA-AKI. The tables, figures, and graphs provided along with the article are very thorough and informative. However, the authors should address the below concerns/comments to improve the quality of the article.

Authors: We thank the Reviewer for their positive consideration of our article and for the useful comments to improve the manuscript.    

  1. Authors should include a couple of sentences in the Introduction highlighting the factors responsible for AKI's pathogenesis due to cardiac surgery. It will help readers who are not experts in the field.

Authors: Yes, thank you. That will certainly help many readers not so much immersed in the field. Some pathogenetic ideas have been included in the Introduction.

  1. Diagnosis of CSA-AKI appears to be the same as the diagnosis of AKI in general. Is there any biomarker or pathophysiological parameters that can directly indicate the involvement of cardiac surgery in AKI?

Authors: Aetiological diagnosis is, indeed, a very relevant aspect of AKI diagnosis. Precisely, we have a research line devoted to this topic, and some articles published in this area. However, these are still at the preclinical level, and none of our studies yet address the differential diagnosis of CSA-AKI, but of other types of AKI. Unfortunately, to our knowledge, there is no information to include in this regard.

  1. Do the authors think that the furosemide stress test could be specific for detecting CSA-AKI? can it not mislead in finding the real cause of kidney failure if considered as specific to CSA-AKI biomarker?

Authors: We totally agree with the Reviewer. The FST is, absolutely, a general test of tubular dysfunction portraying no aetiological information. We do not intend to present it as such, but as a contributor to the general evaluation of the tubular status.

  1. A summary/conclusion section is required to summarize the review in the last.

Authors: Yes, we missed that in the initial version. Thank you for letting us know. We have now included a Conclusions section in the revised manuscript.